# A Novel Saliency-Based Decomposition Strategy for Infrared and Visible Image Fusion

**Biao Qi** [1,*], **Xiaotian Bai** [2], **Wei Wu** [2], **Yu Zhang** [1], **Hengyi Lv** [1] and **Guoning Li** [1]

1    Changchun Institute of Optics, Fine Mechanics and Physics, Chinese Academy of Sciences,
     Changchun 130033, China; zhangyu@ciomp.ac.cn (Y.Z.); lv_hengyi@163.com (H.L.);
     liguoning@ciomp.ac.cn (G.L.)
2    University of Chinese Academy of Sciences, Beijing 100049, China; baixiaotian19@mails.ucas.ac.cn (X.B.);
     wuwei195@mails.ucas.ac.cn (W.W.)
*    Correspondence: qibiao@ciomp.ac.cn

**Abstract:** The image decomposition strategy that extracts salient features from the source image is crucial for image fusion. To this end, we proposed a novel saliency-based decomposition strategy for infrared and visible image fusion. In particular, the latent low-rank representation (LatLRR) and rolling guidance filter (RGF) are together employed to process source images, which is called DLatLRR_RGF. In this method, the source images are first decomposed to salient components and base components based on LatLRR, and the salient components are filtered by RGF. Then, the final base components can be calculated by the difference between the source image and the processed salient components. The fusion rule based on the nuclear-norm and modified spatial frequency is used to fuse the salient components. The base components are fused by the $l_2$-energy minimization model. Finally, the fused image can be obtained by the fused base components and saliency detail components. Multiple groups of experiments on different pairs of infrared and visible images demonstrate that, compared with other state-of-the-art fusion algorithms, our proposed method possesses superior fusion performance from subjective and objective perspectives.

**Keywords:** image fusion; latent low-rank representation; rolling guidance filter; visible image; infrared image

## 1. Introduction

Image fusion is an effective means of image enhancement which is committed to reintegrating an image with more comprehensive information and significant features from multi-sensor images [1]. In this field, the fusion of infrared and visible images is the most representative and widely applied, such as target detection [2], object recognition [3] and surveillance [4].

Visible imaging captures the reflected intensity information, whereas infrared imaging captures the thermal radiation information. The images obtained by two imaging types can provide scene information from complementary aspects [5]. The primary prerequisite of image fusion is that the information conveyed by various sensors is complementary. The chief objective of image fusion is to reintegrate the complementary information from different images of the same scene [6]. In addition, the main challenge is how to extract salient features from the source images and convey them to the fused image as much as possible to improve the visual effects. Some scholars have proposed relevant algorithms to solve this problem, such as the multiscale transform method [7] and the compressive sensing transform method [8].

For the multiscale transform methods, some traditional tools are adopted to decompose source images into one base layer and several detail layers of different scales. Common methods include the discrete wavelet transform (DWT) [9], contourlet transform

(CT) [10], non-subsampled shearlet transform (NSST) [11], rolling guidance shearlet transform (RGST) [12], co-occurrence analysis shearlet transform (CAST) [13], etc. The base layer components that control the global contrast of images can be fused by the averaging fusion rule; nevertheless, the components of detail layers that represent the detail information usually fused by the max-absolute rule. However, the computation complexity of these methods is a little higher because the source images need to be projected to the frequency domain. Henceforth, some methods without the transform are adopted as the processing methods, such as compressive sensing (CS) [14].

Moreover, some scholars utilized pattern spectra and a characteristic scale-saliency-level (CSL) model based on differential area profiles to deliver crisp details of salient features in a strictly edge-preserving manner. In [15], the authors propose a segmentation method by the residuals of morphological opening and closing transforms based on a geodesic metric for the segmentation of complex image scenes. In [16], a CSL model is used to converge to an approximate building footprint representation layer, and its result is a medium abstraction semantic layer used for visual exploration, image information mining and pattern classification.

CS is applied for image fusion, which uses the smaller number of linear data with sparse and compressible attributes to reintegrate and represent image information. The most universal methods mainly include dictionary learning (DL) [17] and sparse representation (SR) [18]. For instance, Ahmed et al. [19] proposed an improved approach for medical image fusion based on sparse representation and the Siamese convolutional neural network. Zhang [20] proposed a visible and infrared image fusion method based on convolution dictionary learning. Moreover, this kind of method does not need any a priori value of the input image during the whole process of representation and can improve the fusion efficiency. However, these methods are still complex and take up more time, especially dictionary learning.

Recently, deep learning-based image fusion algorithms have been researched by more and more scholars, which can be divided into non-end-to-end and end-to-end training ways [21]. Non-end-to-end methods are achieved by deep learning and conventional methods together without training for the first time, such as CNN [22], ResNet-ZCA [23] and VggML [24]. The pre-trained model, as the part of the methods, is used to extract deep features and generate weighted maps. The pre-trained network of the current fusion task cannot be efficient for the others directly. The feature extraction process needs complex pre-processing and post-processing, which results in the increase in computation complexity. Therefore, the end-to-end methods are designed to obtain the salient features via training the network with a lot of images, such as DenseFuse [25], U2Fusion [26], FusionGAN [5] and SwinFusion [27]. The end-to-end fusion methods can learn appropriate parameters adaptively because the framework avoids the complexity of feature extraction in conventional methods. However, most methods cannot design refined fusion rules to extract the deep features due to the simpler network structure. Then, such methods for image fusion tasks can still be improved greatly.

Latent low-rank representation (LatLRR) [28], based on the low-rank representation (LRR), can be used as a clustering analysis tool. Meng et al. [26] proposed a medical image fusion method in which LatLRR is used as the means to extract the salient features of input images. It is robust enough for outliers and noise. Liu et al. [29] adopted LatLRR as the fusion method of visible and infrared images. However, this method lacks spatial consistency and results in artifact effects around the edges. Then, some simple fusion strategies can no longer meet the higher fusion requirements. As a result, the fusion rules are crucial, on which the effect of the fused image depends. Especially for infrared and visible, they are very different in some characteristics with various weather or illumination conditions. It is very important for image fusion to choose an adaptive fusion rule.

Li et al. [30] proposed a fusion method based on the guided filter (GF) transform tool to make up for spatial consistency, which can solve the questions of data distortions and redundancy. However, the performances of the noise removing and edge preservation are

not optimal because only the spatial weights are considered. Then, the rolling guidance filter (RGF) is proposed to solve these defects, which can consider the spatial and range weights together, and the rapid convergence can be achieved through rolling iteration. In [31], Jian et al. realized the image fusion by RGF, and the results are superior to those of the other current edge-preserving filters.

In this paper, we use DLatLRR_RGF as the fusion framework for infrared and visible image fusion, which mainly solves the problem that the salient detail layers decomposed by LatLRR still have some small structure components, and the edges of the base layer also become a bit unclear.

The main contributions of this paper can be briefly summarized as follows:

- The image decomposition method (DLatLRR_RGF) based on LatLRR and RGF is proposed for infrared and visible image fusion. Compared with the fusion framework based on MDLatLRR in [32], which only decomposes the input images to a series of salient layers and one base layer, in this paper, given that the salient detail layers still have a lot of small structural components, RGF is adopted as the processing means to remove the small structural components and recover the edge information. By the way, the base layer has a preponderance of contour information. Finally, the different types of components can be extracted to different layers more delicately, which is conducive to subsequent image fusion processing.
- The projection matrix $L$ of DLatLRR can be calculated in advance during the training phase. Once the projection matrix $L$ is obtained, it can be used to calculate the low-rank coefficients for each image. The size of the image patch needs to be in line with the size of the project matrix $L$; thus, the decomposition means can be adaptive to the image of the arbitrary size.
- The fusion strategies are designed for base components and detail components, respectively. On the one hand, the $\ell_2$ energy minimization model based on the energy information of the base images is adopted to guide the fusion of base components. On the other hand, the nuclear-norm and space frequency are used to calculate the weighted coefficients for every pair of image patches.

The subsequent sections of this paper are organized as follows. In Section 2, we introduce related work. In Section 3, the proposed algorithm is presented in detail. Section 4 shows the experimental settings, results and analysis. Finally, the conclusions are drawn in Section 5.

## 2. Related Works

In this section, for a comprehensive review of some algorithms most relevant to this study, we focus on reviewing the latent low-rank representation and rolling guidance filter.

### 2.1. Latent Low-Rank Representation

LatLRR, as a compressed sensing method, is applied to more and more fields, and image fusion is no exception. In 2010, Liu et al. presented an exploration method based on LRR for the spatial structure of data [28]. However, it is not widely applied because of its two defects:

1. The observed data matrix itself is adopted as the learning dictionary. Therefore, the performance of this method is vulnerable to the observed data, such as insufficient or corrupted [33,34].
2. It considers only the global structure (low-rank representation) information of the observed data, so it cannot retain the local structure (salient features) information as well as possible [34,35].

Subsequently, the authors [29] proposed the LatLRR to address these problems. LatLRR can construct the dictionary based on unobserved latent image data, and the richness and stability of the raw data can ensure the reliability of the dictionary. In other words, even if the input image data are insufficient or damaged, it will not affect the

extraction of salient components. Moreover, LatLRR can comprehensively consider the aspects of the global structure, local structure and sparse noise of input data [28]. LatLRR has better efficiency in extracting the salient features from the corrupted data and strong robustness against the noise and outliers.

Generally, LatLRR is a convex optimization question which can be solved with kernel norm minimization [29], and it can be formulated as

$$\min_{Z,L,E} \|Z\|_* + \|L\|_* + \lambda \|E\|_1 \ \ s.t. \ \ X = XZ + LX + E, \tag{1}$$

where $X$ is the observed matrix of the input data; $Z$, $L$ and $E$ denote the low-rank coefficient matrix, projection matrix of salient coefficients and sparse noise matrix, respectively. $\|\cdot\|_*$ represents the nuclear norm, and $\|\cdot\|_1$ is the $\ell_1$ norm. $\lambda(\lambda > 0)$ is the balance parameter. The question can be solved by the inexact augmented LaGrangian multiplier (IALM) [29] algorithm.

LatLRR can effectively decompose the input image data to low-rank components ($XZ$), the salient components ($LX$) and the sparse noise components ($E$). However, the noise may seriously affect the fusion process and eventually introduce visual artifacts. The noise components can be separated from input images by LatLRR and directly discarded in the proposed algorithm, which is equivalent to the noise removal process and helpful to improving the quality of image fusion [6]. An example of the image decomposition based on LatLRR is shown in Figure 1.

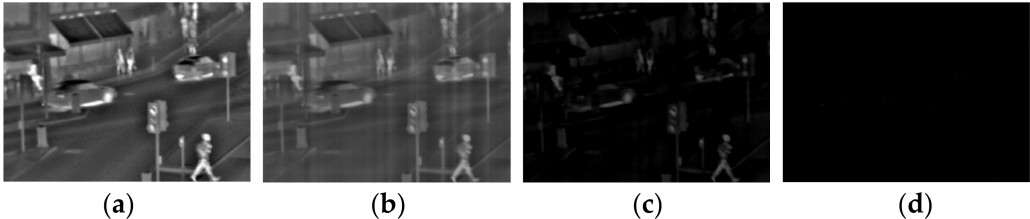

|  (a)  |  (b)  |  (c)  |  (d)  |

**Figure 1.** The decomposition operation based on LatLRR. (**a**) Input image; (**b**) Base part; (**c**) Salient part; (**d**) Sparse noise.

### 2.2. Rolling Guidance Filter

The rolling guidance filter has been one of the most important image smoothing tools since it was proposed. Compared with the other edge-preserving filters, RGF [36], which can achieve the small structure removal and edge information recovery, applies iteration to the filtering process in order to obtain faster convergence. The diagram of image processing based on RGF is shown in Figure 2. RGF consists of two steps: small structure elimination and edge recovery.

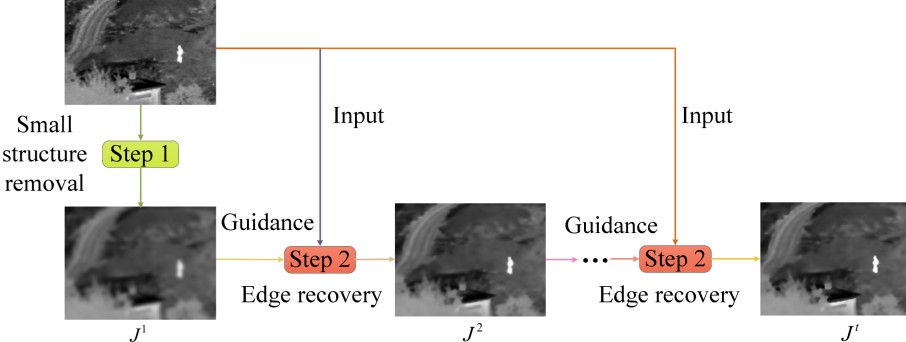

**Figure 2.** Rolling guidance filtering.

2.2.1. Small Structure Elimination

In this process, a Gaussian filter (GF) is applied as the tool for removing the small structure, such as the texture, noise, small target information and so on. The Gaussian filter is defined as follows:

$$G(p) = \frac{1}{K_p} \sum_{q \in N(p)} \exp\left(-\frac{\|p-q\|^2}{2\sigma_s^2}\right) \cdot I(q), \tag{2}$$

$$K_p = \sum_{q \in N(p)} \exp\left(-\frac{\|p-q\|^2}{2\sigma_s^2}\right), \tag{3}$$

where $I$ and $G$ represent the input image and initial guided image, respectively. The parameters $p$ and $q$ are the pixel position index of the image, and $\sigma_s$ denotes the smallest standard deviation of the Gaussian kernel. $K_p$ is used as the normalization, and the neighborhood pixel set at $p$ is defined as $N(p)$. Provided the scales of some components are smaller than $\sigma_s$, GF can remove them completely from the source image during this phase.

2.2.2. Edge Recovery

This step is able to recover the edges of the blurred image $G$ and consists of the joint filter and iteration process. Considering the guided filter with higher computational efficiency and better edge-recovering performance, take it as the joint filtering. The output result of the Gaussian filter is set as the initial guided images $J^1$, and $J^{t+1}$ represents the output of the $t$-th iteration. The process is expressed as follows:

$$J^{t+1} = \frac{1}{K_p} \sum_{q \in N(p)} \exp\left(-\frac{\|p-q\|^2}{2\sigma_s^2} - \frac{\|J^t(p) - J^t(q)\|}{2\sigma_r^2}\right) \cdot I(q), \tag{4}$$

$$K_p = \sum_{q \in N(p)} \exp\left(-\frac{\|p-q\|^2}{2\sigma_s^2} - \frac{\|J^t(p) - J^t(q)\|}{2\sigma_r^2}\right), \tag{5}$$

where $K_p$ is defined as Equation (5), which is also used for normalization. The parameter $\sigma_r$ can control the range weights. $I$ is the input image, as in Equation (2).

## 3. Proposed Algorithm

In this section, the image decomposition method based on LatLRR nested with RGF is first proposed. Two kinds of fusion rules are designed for different layers, respectively.

### 3.1. Pretraining of Projection Matrix L

Based on the introduction in Section 2.2.1, the projection matrix $L$ can be obtained by calculating Equation (1). First, we select five groups of infrared and visible gray images to build a training set, as shown in Figure 3. Second, these images are divided into image patches, whose size is $n \times n$, by the sliding window technique, and the stride of the training window is to achieve the best fusion performance. Moreover, the size of image patches has a certain impact on the fusion results due to the fact that the larger the image size is, the more useful the information it contains. However, the size is so large that the calculation of the projection matrix takes more time. Hence, in our experiments, $n$ is set to sixteen. Third, these image patches are divided into two categories (smooth, detail). The matrix $X$ consists of the two categories according to a certain proportion which depends on the parameter e, and each column of $X$ contains all the pixels of one image patch. The size of $X$ is $M \times N$, where $M$ is the number of image patches, and $N = n \times n$. The size of matrix $L$ only depends on the image patch size, so it can be used to extract the salient features from the input image with an arbitrary size. The setting of parameters $n$ and the stride will be discussed in Section 4.

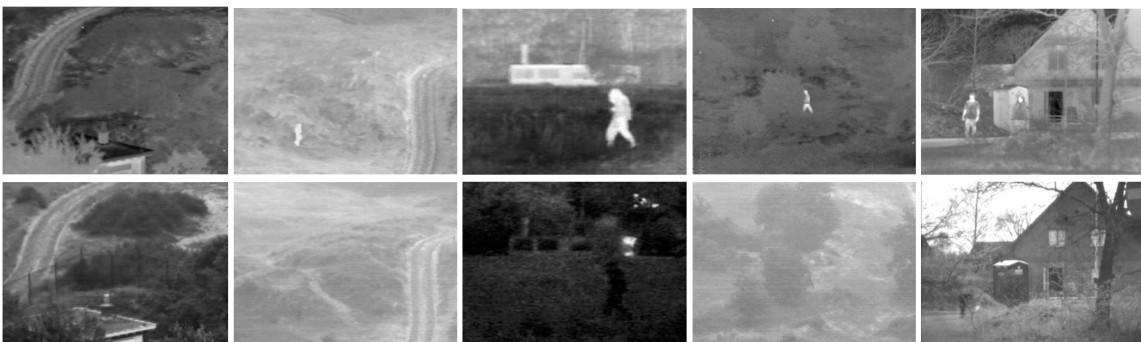

**Figure 3.** Pretraining dataset of five infrared and visible images. The first row presents the infrared images, and the second row is the visible images.

### 3.2. Image Decomposition Based on LatLRR-RGF

Once the projection matrix $L$ is calculated by the LatLRR, we can utilize it to extract the salient detail components, the process of which is illustrated in Figure 4. We can see that the input image is divided into a lot of image patches by means of the sliding window, with some overlap. The size of the sliding window is $n \times n$, which indicates the number of image pixels covered by the window. Then, the window can be moved horizontally and vertically in the stride $s$ at a time.

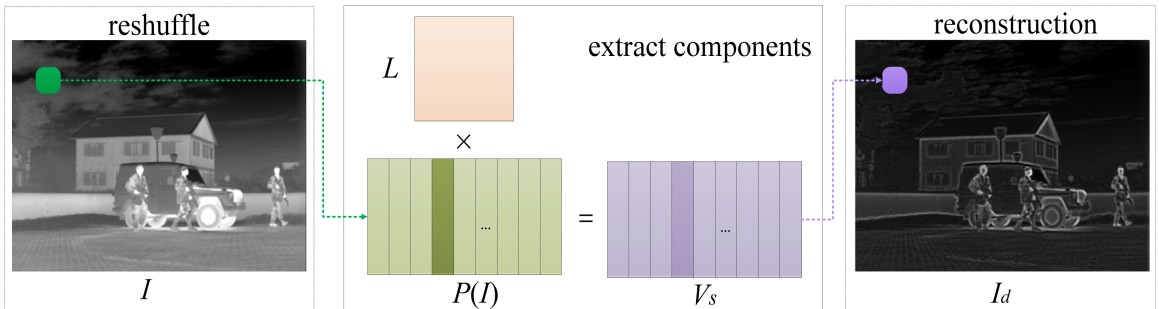

**Figure 4.** The process of salient feature extraction.

These images patches are integrated into a new matrix, each column of which corresponds to some image patch. The salient components can be calculated by the following equations:

$$V_s = L \times P(I), \tag{6}$$

$$I_s = R(V_s), \tag{7}$$

where $I$ and $I_s$ denote the input image and saliency image, respectively. $V_s$ signifies the salient detail components calculated by the projection matrix $L$. $R()$ is the operation of reconstruction from the salient detail components, and the overlapping pixel can be processed by the averaging strategy, namely, calculating the average value of the overlapping pixels in each position. $P(I)$ represents the matrix that consists of the reshuffled patches.

$$I_{s\_rgf} = RGF(I_s), \tag{8}$$

$$I_b = I - I_{s\_rgf}, \tag{9}$$

where $I_b$ denotes the base image, and $I_{s\_rgf}$ is the result processed by RGF. The base image $I_b$ can be acquired by subtracting between the input image $I$ and the salient detail image $I_{s\_rgf}$ smoothed by the RGF.

As shown in Figure 5, the decomposition method can also be applied to every base sub-image. Supposing $r$ represents the highest decomposition level, the value range of the

variable $i$ is defined as $[1, r]$. As a result, $r$ salient detail images and one final base image can be obtained. This framework for multi-level decomposition can be expressed as

$$V_i = L \times P\left(I_b^{i-1}\right), \tag{10}$$

$$I_s^i = R(V_i), \tag{11}$$

$$I_{s\_rgf}^i = RGF\left(I_s^i\right), \tag{12}$$

$$I_b^i = I_b^{i-1} - I_{s\_rgf}^i. \tag{13}$$

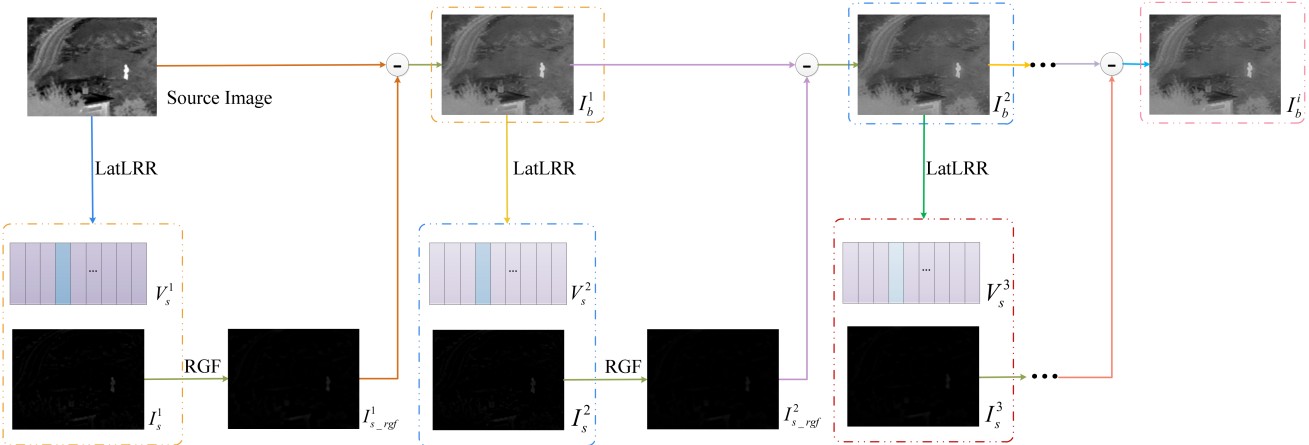

**Figure 5.** The decomposition based on LatLRR nested with RGF.

Finally, the fused image can be reconstructed by the salient detail and base components.

### 3.3. Fusion Method

In this section, the fusion rule for sub-images in different layers is discussed in detail, and the main fusion framework is shown in Figure 6, from which we can know the whole procedures about the image decomposition, image fusion and image reconstruction:

- First of all, the visible images and infrared images can be decomposed by the method based on LatLRR and RGF, which decomposes the input images to a series of salient layers and one base layer. However, given that there are still many small structural components in the salient detail layers, RGF is adopted as a processing tool to remove these small structural components and recover more edge information. By the way, the base layer has a preponderance of contour information. Finally, the different components can be extracted to different layers more delicately, which is conducive to subsequent image fusion processing.
- For the base layer components, the $\ell_2$ energy minimization model based on the energy information of the base images is adopted to guide the fusion. The energy information can reflect the main component mapping, and the weighting map can fuse the infrared and visible base layer components finely.
- For the detail layer components, the nuclear-norm and space frequency are used to calculate the weighted coefficients for every pair of image patches. The space frequency can show the pixel activity of the different detail layers, which can transfer more information to the fused images.

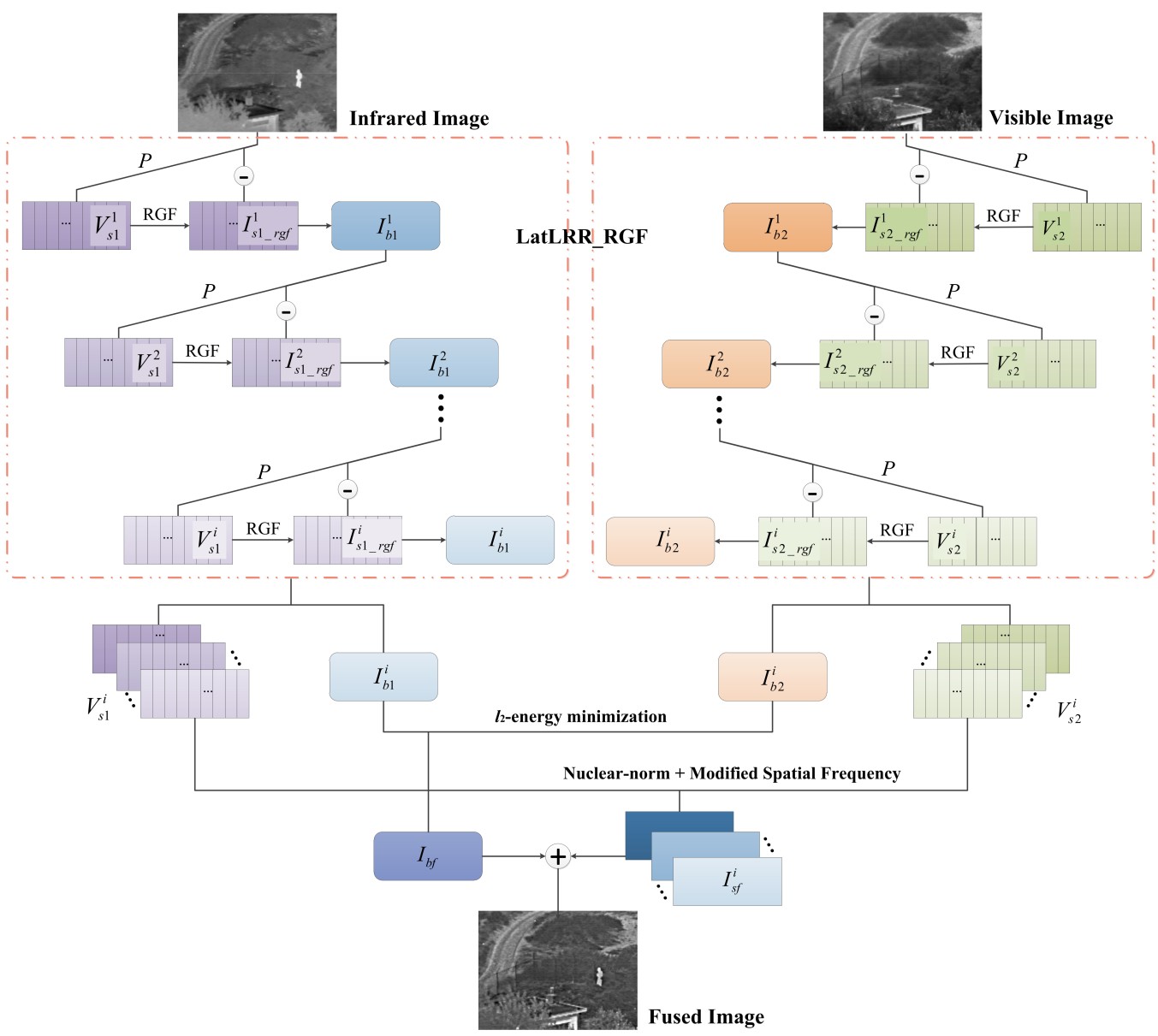

**Figure 6.** The framework of the proposed fusion method.

### 3.3.1. Fusion of Base Components

The base components of the image primarily consist of common features and brightness information. In order to combine the brightness advantages of the infrared image and the information richness of the visible image, the $\ell_2$ energy minimization model based on the energy information of the base images is used to integrate the approximate components of the source images [37]. To implement fusion tasks flexibly, the fusion strategy can adaptively adjust the fusion coefficients according to the local region's energy.

The expression of the fusion model is shown as follows:

$$\arg\min\left\{\left\|I_b(i,j) - I_b^{IR}(i,j)\right\|_2^2 + \lambda\left\|I_b(i,j) - I_b^{VI}(i,j)\right\|_2^2\right\}, \tag{14}$$

where $I_b^{IR}(i,j)$, $I_b^{VI}(i,j)$ and $I_b(i,j)$ represent the base layer components of the infrared image, visible image and fused image, respectively. $\|\cdot\|_2$ denotes the $\ell_2$-norm, and $\lambda$ is the regularization parameter. The data fidelity of this model is $\left\|I_b(i,j) - I_b^{IR}(i,j)\right\|_2^2$, which ensures that the base components of the infrared image can be transferred into the fused

image as much as possible. In addition, $\left\| I_b(i,j) - I_b^{VI}(i,j) \right\|_2^2$ is the regularization term in which the base components of the visible image are introduced, and it is beneficial to improve the sensitive observation range of human eye vision. The Euler–LaGrange function of Equation (14) can be calculated, and the result is shown as follows:

$$\left( I_b(i,j) - I_b^{IR}(i,j) \right) + \lambda \left( I_b(i,j) - I_b^{VI}(i,j) \right) = 0, \tag{15}$$

In order to express the contribution from the source image to the fused image, we introduce the weighted parameters $\omega_1$ and $\omega_2$. Thus, the solution to Equation (14) is that

$$I_b(i,j) = \omega_1 I_b^{IR}(i,j) + \omega_2 I_b^{VI}(i,j)$$

$$s.t. \ \omega_1 = \frac{1}{1+\lambda}, \ \omega_2 = \frac{\lambda}{1+\lambda}, \tag{16}$$

where the weighted parameters mainly depend on the regularization parameter $\lambda$ that can influence the final fusion effect. Given $\lambda$ with a fixed constant, the information in two different types of spectra images cannot be fully extracted. Thus, the value of $\lambda$ should be a variable that can vary with the gray features of the input images. Moreover, the parameter $\lambda$ is not only a correction parameter but can also express the local details of the visible image. In order to improve the image contrast, we adopt the local standard deviation (LSD) [38] as the adaptive regularization parameter, which can be defined as follows:

$$\lambda(i,j) = \sqrt{\frac{1}{MN} \sum_{m \in M} \sum_{n \in N} \left[ I_b^{VI}(i+m, j+n) - \overline{I_b^{VI}}(i,j) \right]^2}, \tag{17}$$

$$\overline{I_b^{VI}}(i,j) = \frac{1}{MN} \sum_{m \in M} \sum_{n \in N} \left| I_b^{VI}(i+m, j+n) \right|, \tag{18}$$

where $M$ and $N$ represent the number of the pixels in the region, the size of which can be $3 \times 3$ or $5 \times 5$. $\overline{I_b^{VI}}(i,j)$ denotes the average gray of all pixels in the local area, and it can be calculated by Equation (18).

As a result, the variable $\lambda$ can compensate well for the data fidelity term in $\ell_2$ energy minimization and retain the region in the images that is the main one or sensitive for human vision.

### 3.3.2. Fusion of Detail Components

The salient detail components of the source image mainly consist of saliency features and structural information [39]. Thus, the fusion strategy for detail components needs to be designed more carefully. The fusion rule in this paper is shown in Figure 7.

The nuclear-norm can represent the structural information of each image patch well, and the spatial frequency (*SF*) can sensitively indicate the pixel activity levels. Based on the analysis above, in our method, we combine the nuclear-norm and *SF* to describe the structural information and salient information of the detail images, respectively. Both of them are used to calculate the weighted coefficients for every pair of image patches.

In Figure 7, $i$ represents the decomposition level. $V_{sk}^{i,j}$ ($k = 1, 2$) and $V_{sf}^{i,j}$ denote the $j$-th column of each detail component matrix $V_{sk}^i$ and fused detail component matrix $V_{sf}^i$.

The nuclear-norm for each column is calculated as follows:

$$\hat{w}_{sk}^{i,j} = \left\| R\left( V_{sk}^{i,j} \right) \right\|_*, \tag{19}$$

where $\|\cdot\|_*$ denotes the nuclear-norm.

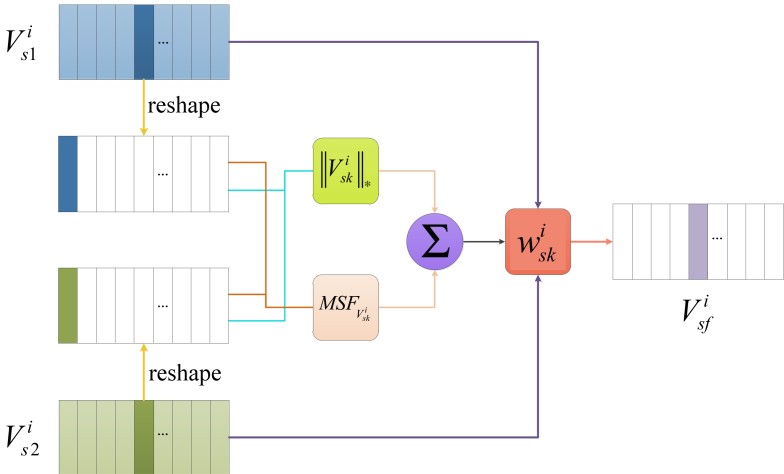

**Figure 7.** Process of fusing the salient detail components.

Based on the traditional *SF*, our method introduces the main diagonal *SF* and the secondary diagonal *SF*, which can acquire more direction and detail information. The modified *SF* (*MSF*) is defined as

$$MSF = \sqrt{(RF)^2 + (CF)^2 + (MDF)^2 + (SDF)^2}, \tag{20}$$

$$RF = \sqrt{\frac{1}{n \times n} \sum_{x=1}^{n} \sum_{y=2}^{n} [d(x,y) - d(x,y-1)]^2}, \tag{21}$$

$$CF = \sqrt{\frac{1}{n \times n} \sum_{x=2}^{n} \sum_{y=1}^{n} [d(x,y) - d(x-1,y)]^2}, \tag{22}$$

$$MDF = \sqrt{\frac{1}{n \times n} \sum_{x=2}^{n} \sum_{y=2}^{n} [d(x,y) - d(x-1,y-1)]^2}, \tag{23}$$

$$SDF = \sqrt{\frac{1}{n \times n} \sum_{x=2}^{n} \sum_{y=1}^{n-1} [d(x,y) - d(x-1,y+1)]^2}, \tag{24}$$

where *MSF* is the modified *SF*, and *RF*, *CF*, *MDF* and *SDF* represent the row, column, main diagonal and secondary diagonal *SF*, respectively. We calculate the *MSF* of every patch, and the size of the statistic area is $n \times n$.

Based on the above analysis, we can obtain the final fusion coefficients $w_{sk}^i$ by Equation (25).

$$w_{sk}^{i,j} = \frac{MSF_{V_{sk}^{i,j}} \cdot \hat{w}_{sk}^{i,j}}{\sum\limits_{k=1}^{2} MSF_{V_{sk}^{i,j}} \cdot \hat{w}_{sk}^{i,j}}, \tag{25}$$

where $MSF_{V_{sk}^{i,j}}$ is the modified SF for each patch obtained by Equation (20), and $\hat{w}_{sk}^{i,j}$ can be calculated by Equation (19). Then, the fused detail components $V_{sf}^{i,j}$ are calculated by weighted detail components $V_{sk}^{i,j}$, as shown in Equation (26):

$$V_{sf}^{i,j} = \sum_{k=1}^{2} w_{sk}^{i,j} \cdot V_{sk}^{i,j}. \tag{26}$$

Equation (26) is applied to every pair of detail component matrices $V_{sk}^i$. ($i = 1, 2, \ldots , r$). Finally, the fused image patches can be reshaped by Equation (27):

$$I_{sf}^i = R\left(V_{sf}^i\right). \tag{27}$$

### 3.4. Reconstruction

The fused base components and fused salient components are together used to reconstruct the fused image by adding the operation, which is shown as

$$I_f(x,y) = I_{bf}(x,y) + \sum_{i=1}^r I_{sf}^i(x,y), \tag{28}$$

where $I_f(x,y)$ denotes the fused image. $I_{bf}(x,y)$ and $I_{sf}^i(x,y)$ are the fused base components and salient components, respectively.

## 4. Experimental Results and Analysis

In the following, the fusion performance of the proposed algorithm is discussed and compared with the other state-of-the-art methods from subjective and objective perspectives.

### 4.1. Experimental Setting

To demonstrate the superiority of the proposed algorithm, 10 pairs of infrared and visible images in different scenes are used as the test data in Figure 8 from TNO (Available online: https://figshare.com/articles/TN_Image_Fusion_Dataset/1008029, accessed on 1 May 2021). The testing set consists of 10 pairs of source images, where the first four pairs of images are used to test the fusion performance in the case of target hiding. The light difference between the infrared and visible images is big enough in the fifth and sixth pairs of images, which can also be utilized as a testing input to verify the fusion effect. The last three pairs of images are taken under the low illumination that is also very demanding for the fusion method with higher adaptability. The ten pair consists of remote sensing images.

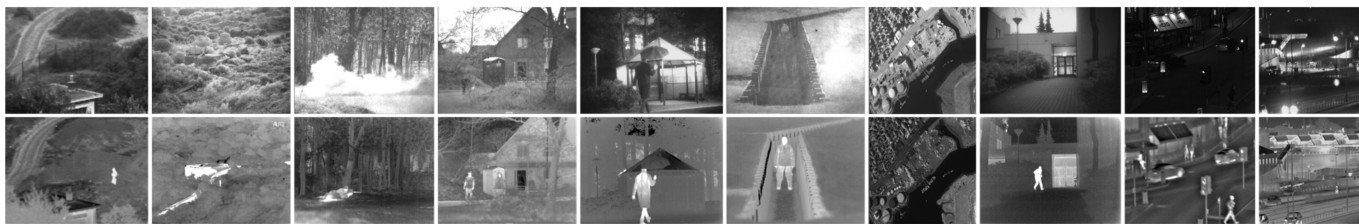

**Figure 8.** Ten pairs of source images. The top row shows the visible images, and the second row contains the infrared images.

To test the superiority and feasibility of the algorithms, we chose the comparison methods from the various aspects, such as the methods focused on the image transformation, the methods based on sparse representation or filtering methods, the deep learning methods and so on. The comparison methods in this paper mainly include curvelet transform (CVT) [40], complex wavelet transform (CWT) [41], guided filtering fusion (GFF) [38], gradient transform (GTF) [42], hybrid multi-scale decomposition fusion (HMSD) [43], Laplacian pyramid with sparse representation (LP_SR) [44], ratio pyramid (RP) [45], fusion based on median filtering (TSF) [46], the weighted least square optimization-based method (WLS) [47], anisotropic diffusion fusion (ADF) [48], U2fusion [26] and Densefuse [25]. The implementation of these compared methods is publicly available, the parameters of which are strictly in accord with the original papers. The experiments were conducted on a desktop with 3.6 GHz Intel CPU i7-6850K, GeForce GTX 1070 Ti and 32 GB memory.

As for the objective metrics, to facilitate a comparison with other existing algorithms, eight performance metrics are chosen to compare the proposed method and the other

existing fusion methods, such as the average gradient (AVG) [49], entropy (EN) [50], mutual information (MI) [17], quality of visual information (Qabf) [51], spatial frequency (SF) [39], standard deviation (SD) [43], structural similarity (SSIM) [52] and sum of the correlations of differences (SCD) [53]. The fusion performance is proportional to the increase in these metric values. Moreover, the running time of different methods is also regarded as a metric reference.

### 4.2. Comparison of the Fusion Effect with and without RGF

In order to prove the recovery effect of the algorithm RGF, we select the image "Butterfly" to confirm the decomposition performance and the image pair "Kayak" to test the final fusion effect with and without the RGF algorithm, respectively. The results are shown in Figures 9 and 10 and Table 1 from subjective and objective perspectives. In Figure 9, the edges of picture (c) are clearer and sharper than those of picture (b). Of course, the pictures (d) and (e) of L2 are the same. It is obvious that RGF can effectively recover the damaged edges in the process of the salient component extraction.

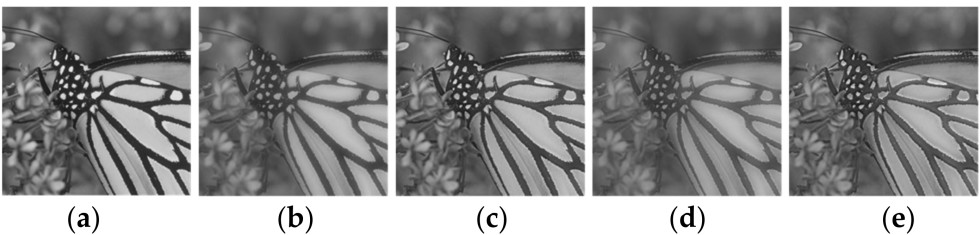

|  (a)  |  (b)  |  (c)  |  (d)  |  (e)  |

**Figure 9.** The comparison of the decomposition effect of the LatLRR method with and without RGF. (**a**) is the original image. (**b**,**c**) are the base layers of L1 without and with RGF. (**d**,**e**) are the base layers of L2 without and with RGF.

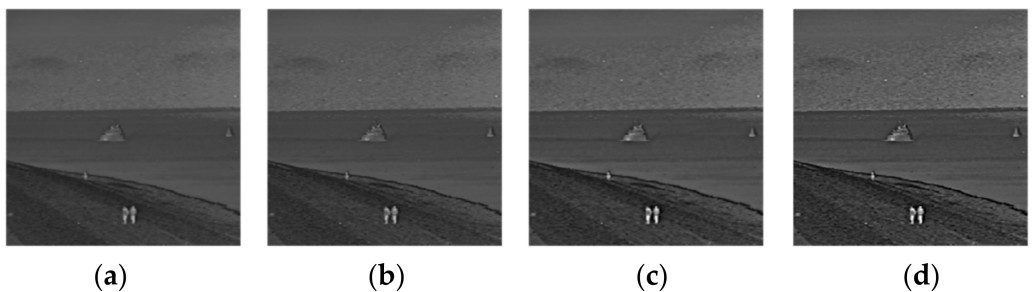

|  (a)  |  (b)  |  (c)  |  (d)  |

**Figure 10.** The comparison of the fusion effect of the LatLRR method with and without RGF for the "Kayak" image pair. (**a**,**b**) are the decomposition of L1 without and with RGF. (**c**,**d**) are the decomposition of L2 without and with RGF.

**Table 1.** The quality metrics of the fused image for the "Kayak" image pair.

| Metrics | AVG | EN | MI | $Q_{abf}$ | SCD | SF | SD |
|---|---|---|---|---|---|---|---|
| L1 (without RGF) | 1.87330 | 5.94780 | 11.89561 | 0.61660 | 1.74054 | 4.06320 | 19.69617 |
| L1 (with RGF) | 2.49603 | 6.01317 | 12.02634 | 0.68572 | 1.75062 | 5.34973 | 20.18842 |
| L2 (without RGF) | 3.05346 | 6.13420 | 12.26839 | 0.72052 | 1.76770 | 6.62708 | 21.94118 |
| L2 (with RGF) | 4.43871 | 6.24749 | 12.49498 | 0.63024 | 1.76299 | 9.41302 | 23.37920 |

In Figure 10, when the decomposition level is set at 1, (a) and (b) are the results of LatLRR without RGF and with RGF recovery. There is much more detail information in (b) than in (a), and the figure is clearer, especially the contour of the steamboat. Of course, the conclusion can be delivered from the first two rows of Table 1. Moreover, (c) and (d) are the results of decomposition level 2, in which the effect of the recovery of the RGF method is also very distinct. However, for the metric Qabf, the value decreases slightly when the

decomposition level increases to 2. The SCD of decomposition level 2 with RGF recovery is lower than that without RGF recovery.

Based on the above analysis, the recovery method based on RGF is beneficial to improving the visual effect of the fused images, and the images have higher contrast and clearer edge information.

### 4.3. Projection Matrix L

There are two key parameters for the calculation of the projection matrix *L*: the patch size *n* and the threshold *e* for classification, which will be discussed in detail.

#### 4.3.1. The Patch Size n

In order to understand the impact of different patch sizes on the fusion efficiency and results, the patches with sizes of 4, 8 and 16 are used for comparison experiments. Figures 11 and 12 show the fusion results and performance of different projection matrices *L* with various sizes on the image pair "Steamboat", among which the stride is set to 1, and the decomposition level is 2.

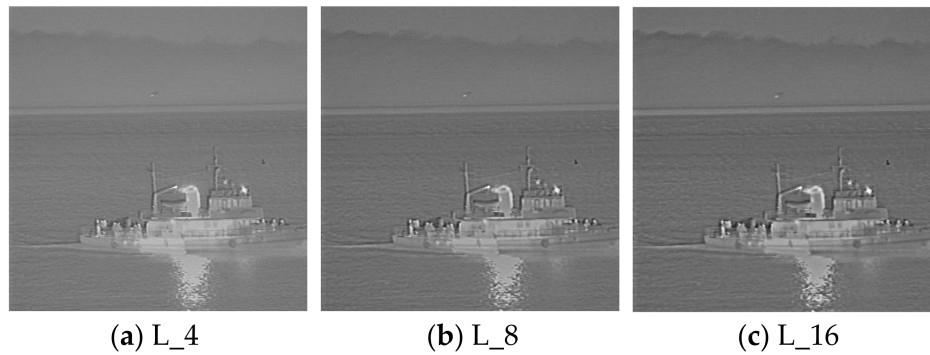

(**a**) L_4  (**b**) L_8  (**c**) L_16

**Figure 11.** The fusion results of different projection matrices *L* on the image pair "Steamboat".

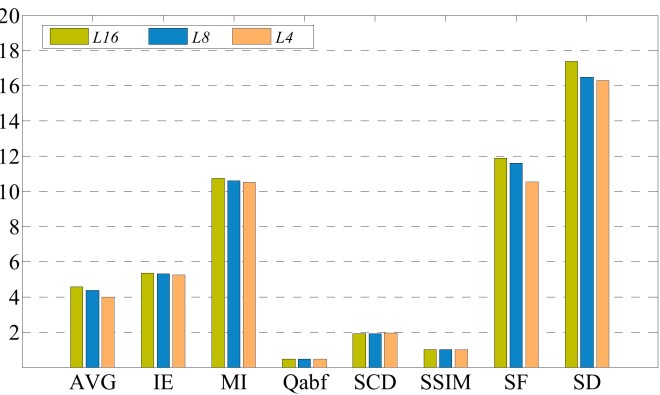

**Figure 12.** The fusion performance of different projection matrices *L*.

In Figure 11, the fusion image with an image block size of 16 has significantly better visual effects than those with other smaller sizes, and there is more detail information in the figure (c). Moreover, the eight metrics in Figure 12 are chosen to further evaluate the fusion images. Except for the two metrics SCD and SSIM, the other metrics of the projection L_16 are the highest. Thus, from an objective perspective, the fused image based on the projection L_16 retains much more edge information and is clearer. However, SCD and SSIM are also very close to the optimal values. As a result, with the increase in the patch size *n*, the fusion effects have been gradually improved, but the computation cost greatly increases. In summary, this paper selects the patch size of 16 to obtain a better fusion effect.

4.3.2. The Threshold *e*

The threshold *e* is used to balance the number of smooth patches and detail patches, and the classification based on the metric *EN* is defined as

$$EN = -\sum_{l=0}^{L-1} P_l \ln(P_l), \tag{29}$$

where *L* denotes the maximum gray level of the image, and the statistical probability of pixels in the image patches at the grayscale level *l* is represented by $P_l$. The larger *EN* indicates that there is more detail information in the image patch. The classification strategy is defined as

$$C(P) = \begin{cases} smooth & EN(P) \leq e \\ detail & others \end{cases}, \tag{30}$$

where *P* denotes the image patch, and *e* represents the threshold value for dividing the patches into smooth or detail segments.

These image patches are divided into two categories: the smooth and the detail. Then, randomly choose the smooth and detail patches from the two categories to generate the input matrix *X*. Thus, the number of smooth and detail patches in the training matrix *X* is important enough for the projection matrix *L*. The ratio of detail and smooth is set at 0.5.

In order to objectively reflect the influence with the change in the threshold *e* and obtain the optimal value, we also carried out the comparative experiments on five image pairs. The size of the image patches is set to 16, and the decomposition level is 2. The results of the image pair of "Steamboat" are shown in Figure 13.

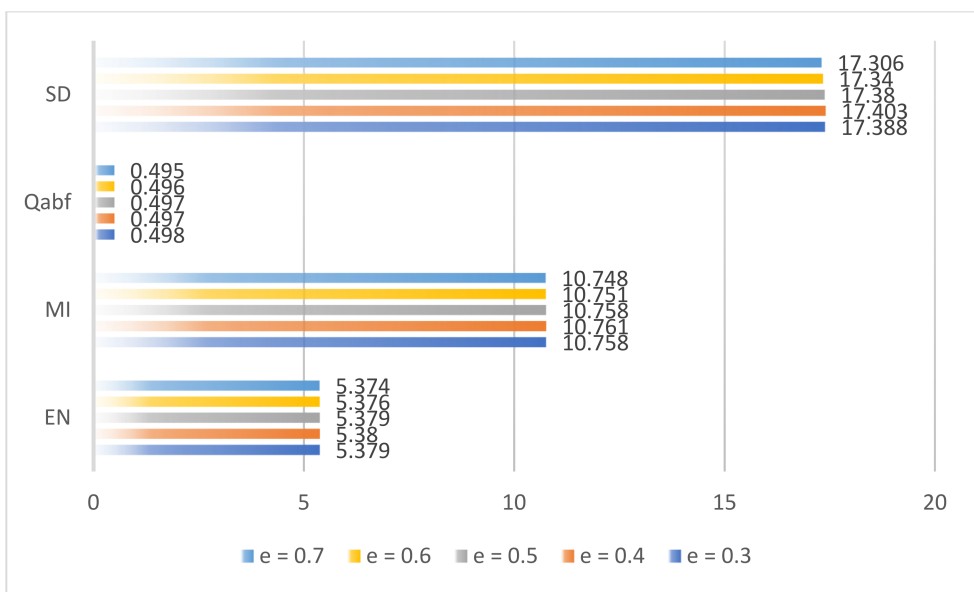

**Figure 13.** Average fusion results of the different threshold *e*.

The experiment results in Figure 13 indicate that with the increase in the threshold *e*, the metrics also fluctuate. When *e* = 0.4, the fusion metrics are the best, other than the $Q_{abf}$, but $Q_{abf}$ is very close to the optimal value. Thus, we choose the threshold *e* = 0.4 in this paper.

*4.4. Decomposition Methods Compared*

The performance of the image decomposition based on the LatLRR_RGF method can be proved by the following experiments. RGST is adopted as the compared method. In Figure 14, the base layer components decomposed by the proposed method are more abundant, such as (e) and (f). Moreover, (e) and (f) have clearer edges than (a) and (b) obtained by RGST, and the ability of edge recovery is better than that of RGST. Especially,

the second level base image (b) has become a little blurry. Comparing the picture (c) with (g), we can know that the salient components extracted by the projection *L* are also subtler and richer, and the intensity information is extracted together. However, (c) and (d) obtained by RGST contain more edge information, and the contour is also clearer. The green box in figures (c) and (d) lacks flower information.

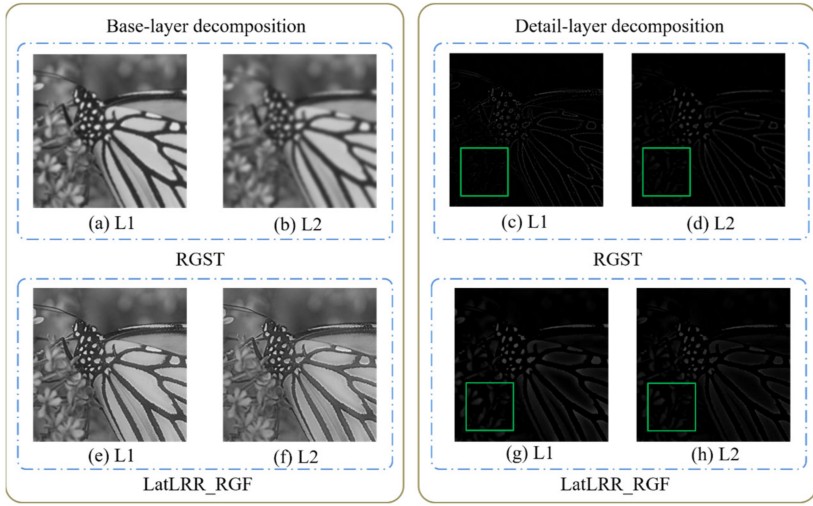

**Figure 14.** The image decomposition experiments. The first row is based on the RGST method, and the second row denotes the LatLRR_RGF method.

Generally, the difference between LatLRR_RGF and RGST is due to their own decomposition principle. The LatLRR_RGF method is usually used to extract the salient components. Thus, the pictures (g) and (h) have more information, including the edge and intensity information. As for the RGST method, it can perform the corresponding decomposition according to the edge scale information of the source image. So, the main energy information is kept in the base layer, and the edge information is shown in the detail layer. However, the more abundant the sub-layer images are, the better the fusion effect is. The proposed decomposition method is competent for this work.

*4.5. Fusion Rules Compared*

The fusion effect mainly depends on the fusion rules, except for image decomposition methods. An appropriate fusion rule is useful to improving the visual effect. In addition, some classic fusion rules have not adapted to all the different scenes. We choose the average method as the base component fusion rule and the absolute-max method as the detail component fusion rule to compare with our fusion methods, respectively. The experiment results are shown in Figures 15 and 16. Figure 15 is the subjective fusion effect comparison experiment for base and detail components. For the base layer fusion rule comparison, the $l_2$ energy minimization rule can process the base components more intelligently instead of the unified averaging operation for all pixels. For example, comparing picture (a) with (b), the image in the box is zoomed in, in which the lamplight fused by the $l_2$ energy minimization rule is more in line with the human visual effect due to the brighter light. Moreover, the proposed method takes the intensity difference of the two kinds of images fully into account, and the final fused effect is better than the average fusion rule. It is more apparent that the metric values of the fused image are larger than the other classic fusion rules in Figure 16 from the objective perspective, especially the AVG, SF, SD and SCD.

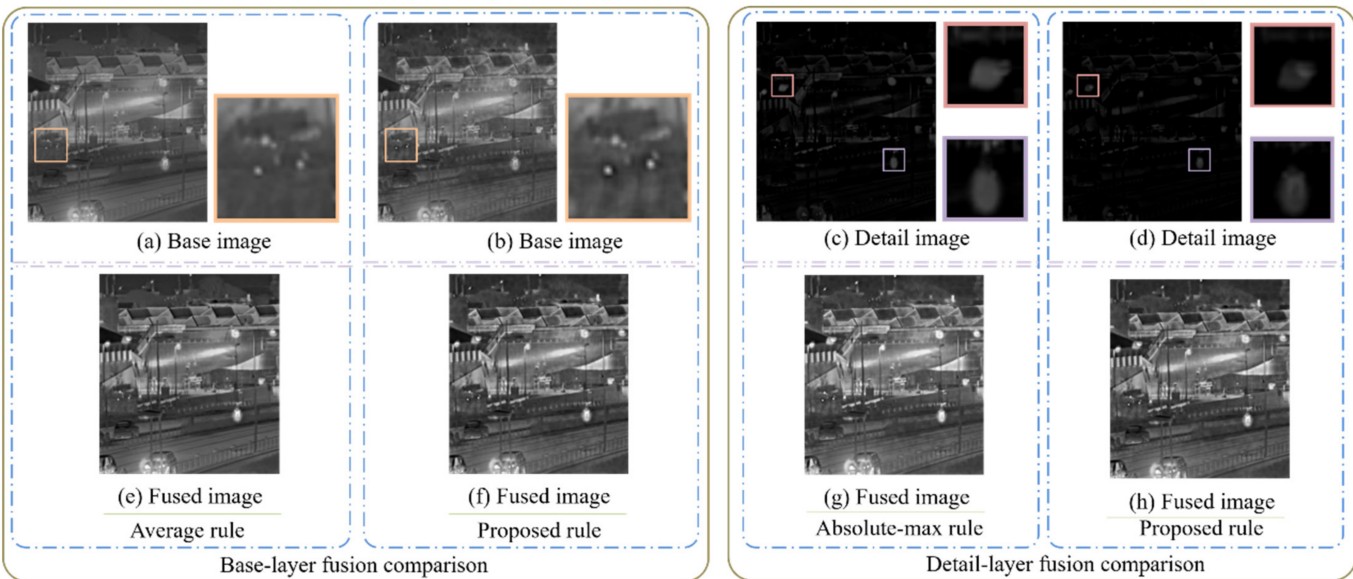

**Figure 15.** Performance comparison of different fusion rules on the image pair "Airplane".

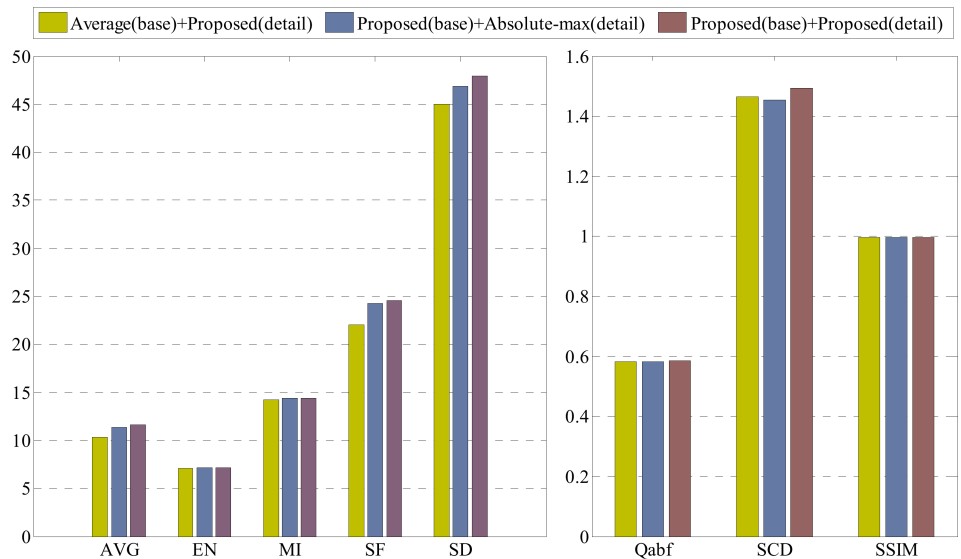

**Figure 16.** Metrics comparison of different fusion rules on the image pair "Airplane".

For the detail layer components, the paper adopted the nuclear-norm and modified the spatial frequency together to guide these image blocks to fuse. In Figure 15, the pictures (c) and (d) are detail images obtained by the absolute-max rule and the proposed rule. We can know that the fusion components based on the proposed method are finer. For example, the gap between the two legs and the contour of the lamp in the light is clearer in picture (d). However, they are integrated into a mass in picture (c). The fusion metrics in Figure 16 show that the proposed fusion rule has a better performance than the absolute-max rule.

The fusion rules proposed in this paper can improve the fusion effect availably, which has been proven from objective and subjective perspectives.

### 4.6. Subjective Evaluation

The subjective evaluation for the fusion of infrared and visible images mainly depends on the visual effect of fused images. The representative image pair "Kaptein" is selected for detail analysis, and the results are shown in Figure 17. The results are obtained by 12 existing methods, in which the parameters take default values and our proposed algorithm that uses the projection matrix L16, and the decomposition levels are r (1 to 2).

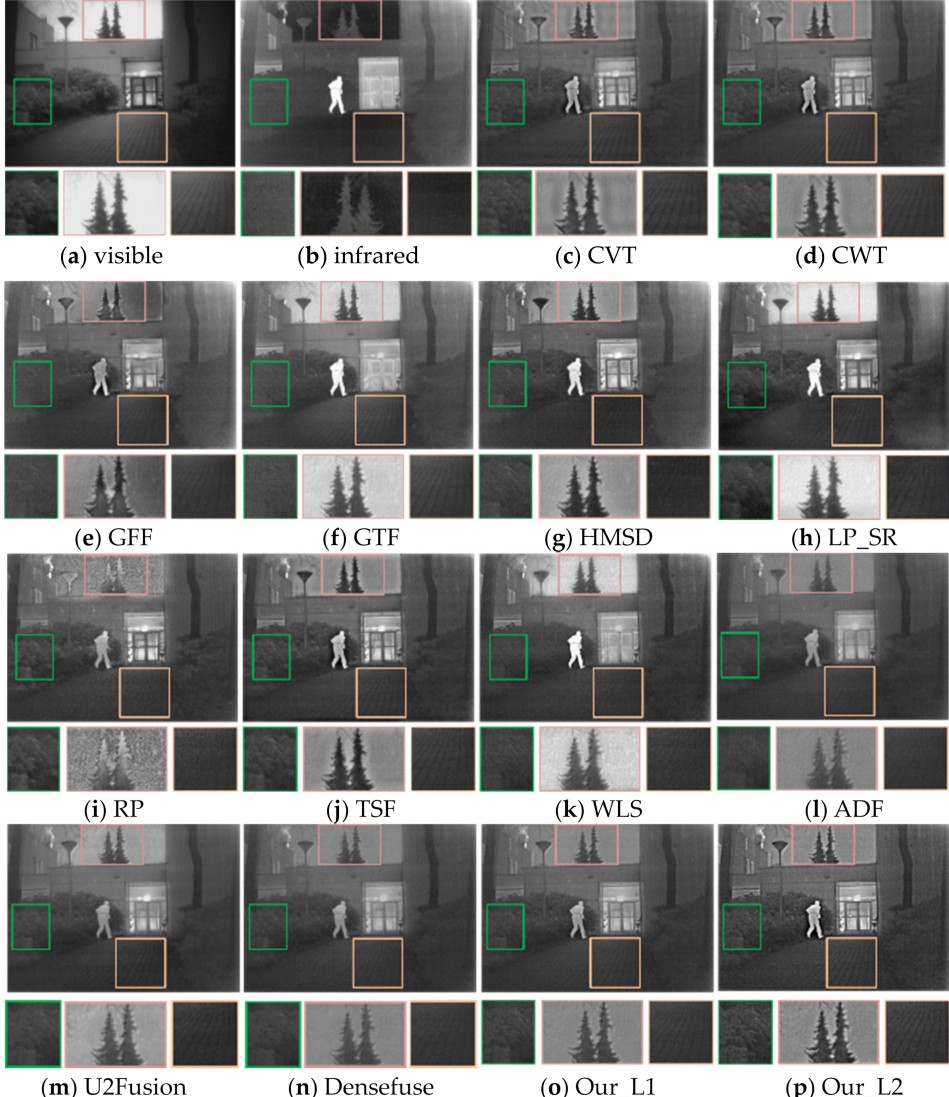

**Figure 17.** Performance comparison of different fusion methods on the image pair "Kaptein". And images in the squares are the localized magnification.

Three parts in the images are enlarged for detail comparison, namely, the trees, the bushes and the ground. It can be seen from the enlarged pictures of the fusion results obtained by various methods that the proposed method with decomposition level 2 owns the more stable fusion performance. The contrast of (p) is high enough, and the quantity of information is adequate. Furthermore, there is no noise, as in the RP and WLS methods. The GFF, GTF, WLS and ADF methods cannot ensure the clarity of fused images, such as bushes in the green boxes. The fused images by the CVT and CWT methods have an outline around some big smooth region such as the left and right sides of the tree at the top of the images. The HMSD, LP_SR and TSF methods have a relatively better fusion effect, but the image contrast is not as good as that of Our_L2. The performance of the deep learning methods, U2Fusion and Densefuse, is at a moderate level, and U2Fusion is a bit better than Densefuse in terms of the image contrast, such as the texture of the ground. Then, the proposed method with two decomposition levels can integrate more information into the final results than only one decomposition level. In summary, our method with decomposition level 2 can retain more detail information and has a great advantage in terms of contrast and clarity.

Figure 18 shows the fusion results of different methods for the remote sensing images. For this experiment, both infrared and visible images contain plenty of useful information.

All the methods can finish the fusion task well. However, the letters in the bottom left corner of the "f", "j", "m", "n" and "o" are not as clear as others. The contrast of "h", "i" and "l" is a little lower than that of "g" and Our_L2. The proposed method can also obtain a good effect on the human vision.

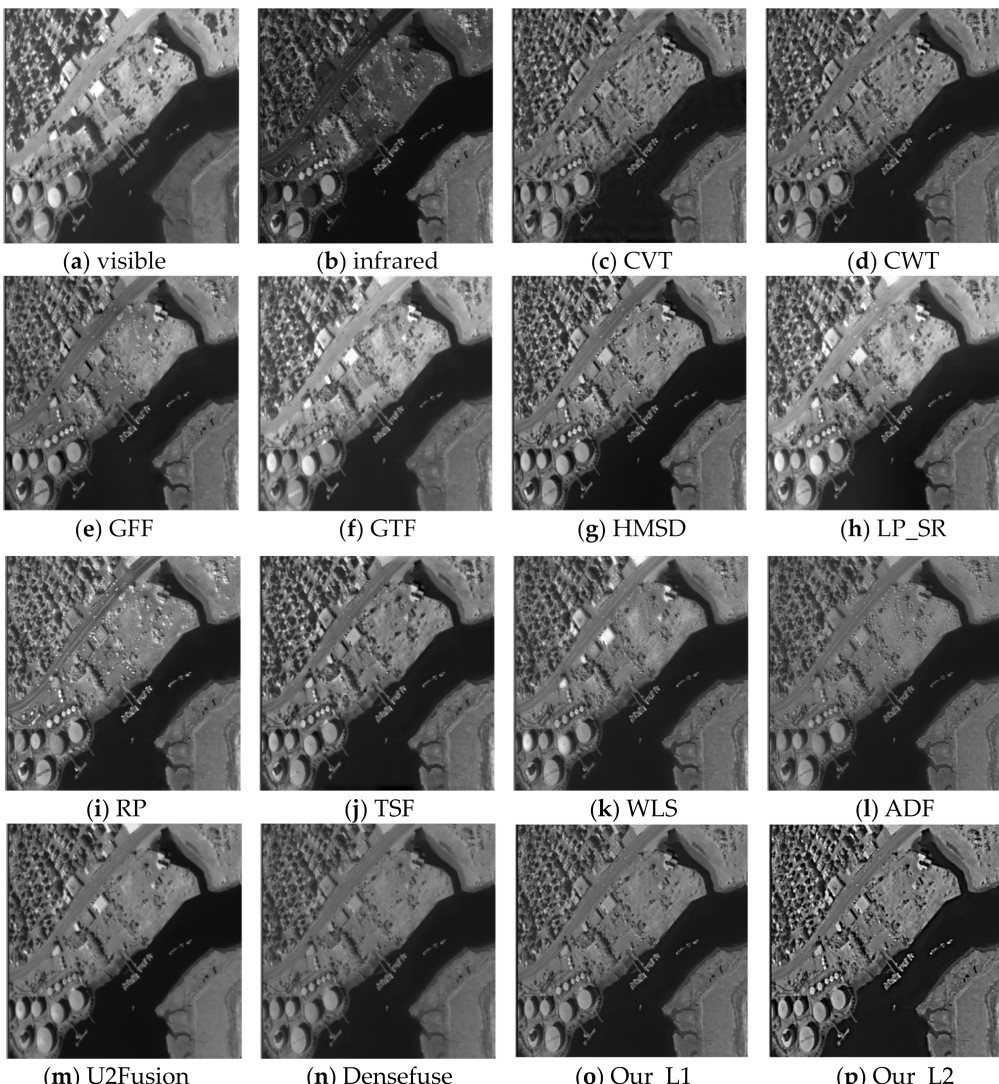

(**a**) visible    (**b**) infrared    (**c**) CVT    (**d**) CWT

(**e**) GFF    (**f**) GTF    (**g**) HMSD    (**h**) LP_SR

(**i**) RP    (**j**) TSF    (**k**) WLS    (**l**) ADF

(**m**) U2Fusion    (**n**) Densefuse    (**o**) Our_L1    (**p**) Our_L2

**Figure 18.** Performance comparison of different fusion methods on the image pair "Remote Sensing".

To verify the above analysis further, Figure 19 shows the results of nine group experiments on infrared and visible images in different scenes. Among these, the first two rows represent source visible image and infrared images, and (c)–(p) represent the results of CVT, CWT, GFF, GTF, HMSD, LP_SR, RP, TSF, WLS, ADF, U2Fusion, Densefuse and the proposed method of the decomposition level 1 (L1) and level 2 (L2), respectively. Whether it is the selected comparison method or the proposed method, the basic fusion task of infrared and visible images can be completed, except for method (e). In the second and sixth experiments, the infrared targets are very unclear in the fused images obtained by method (e), and even the hot targets of the men are lost in the fourth experiment. The rest of results obtained by method (e) are acceptable. Thus, method (e) cannot be suitable for every scene.

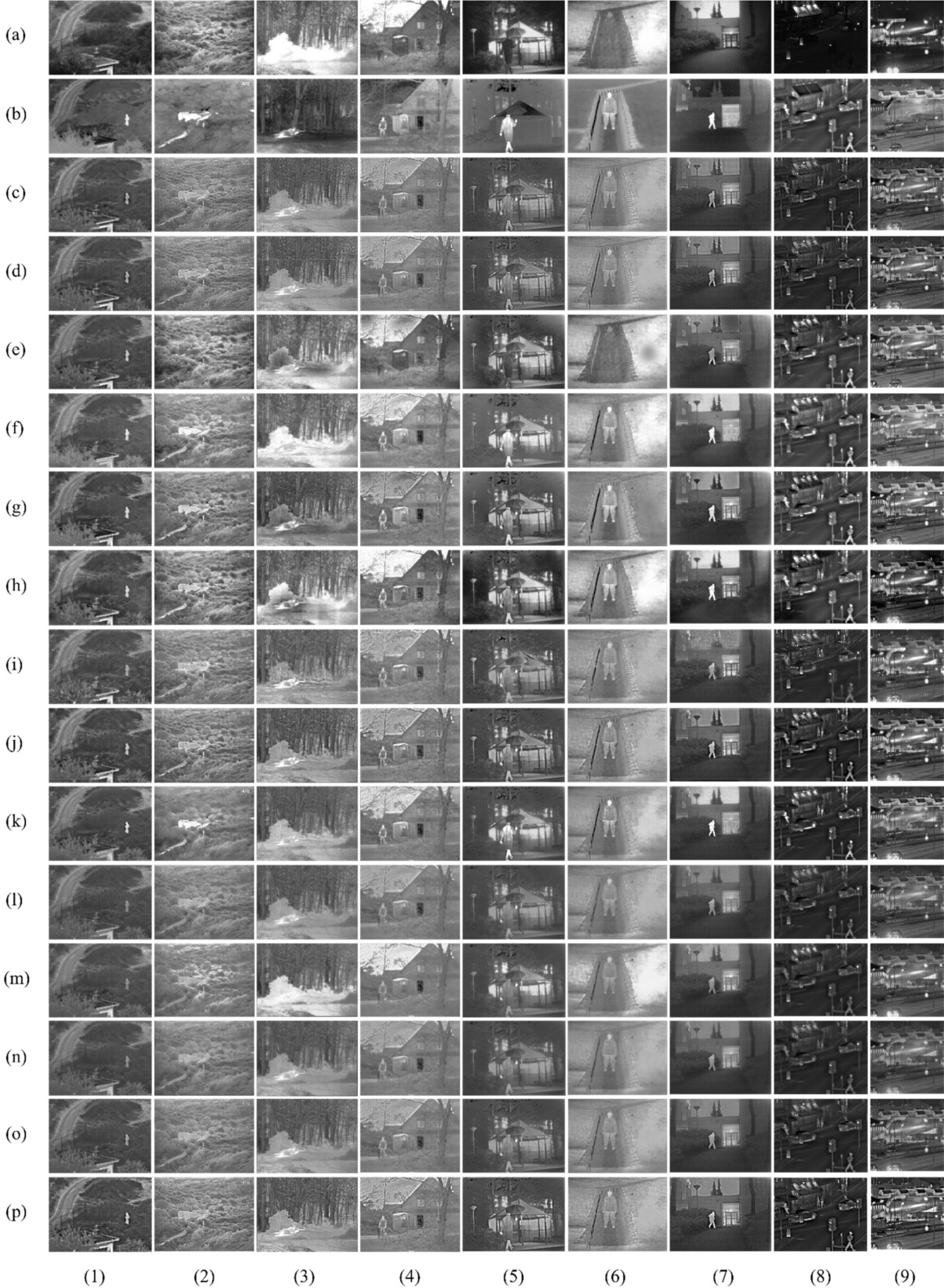

**Figure 19.** Performance comparison of different fusion methods on nine image pairs. (**a**,**b**) are the source images of visible and infrared. (**c**–**p**) are the results of CVT, CWT, GFF, GTF, HMSD, LP_SR, RP, TSF, WLS, ADF, U2Fusion, DenseFuse, Our_L1 and Our_L2. And (**1**–**9**) are experimental group numbers.

The contrast of the fused images from methods (k) and (l) is low, and they are relatively blurred so that the visual effect is not very good. The fused image of method (i) is closer to the corresponding visible image—for example, in the eighth experiment, the detail information of the visible image is more sufficient, but the infrared information cannot be fully transmitted to the fused image, as in experiment (5). In addition, there is a lot of noise in the seventh experiment image based on method (i).

Although the infrared targets of the fusion images from the methods (f), (g), (h) and (k) are salient, some visible detail information is lost—for instance, in the fifth experiment, the texture information of the bushes in the fused image from the methods (f), (g) and (k) is insufficient, and in the sixth group of the experiment, the man's eyes are missing. The soldiers in methods (f) and (k) have been sheltered from the smoke.

According to the fusion results in Figure 19, the schematic diagram of the fusion effect can be summarized by Figure 20 from the perspectives of the image clarity, contrast, visual effect and algorithm stability.

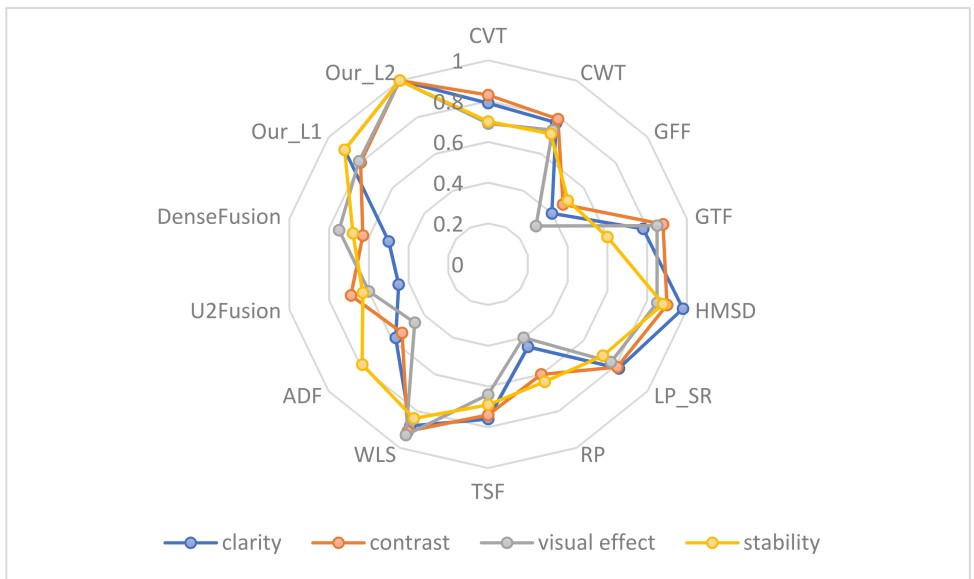

**Figure 20.** The schematic diagram of the fusion effect based on subjective evaluation.

The overall brightness of the images based on methods (c) and (d) are low, resulting in the poor visual effect. The fusion performance of method (e) is unstable for balancing the weights in the fused images between visible and infrared images, such as the ninth and the second group experiments. Moreover, the fusion effect of method (g) is outstanding besides the seventh experiment because the grid texture information of the ground is missing. The fused image based on method (h) exists as an artifact in the third group. The results obtained by method (i) have some noise, especially in the seventh and eighth experiments. The method (j) is relatively stable, although the results also lost some detail information, such as the man's eyes in sixth group. The method (m) integrated more visible information into the results, but for the third and ninth group images, it lost the target information of the infrared images because the visible images have big and bright regions. The fusion performance of the Densefuse method is a little better than that of U2Fusion.

For our method, there is more salient information in the fusion results of the decomposition level 2 than in those of level 1. The fusion images in the last row in Figure 19 show that the contrast and clarity are optimal, and the visual quality of the proposed method is better. From Figure 20, we can know that the fused images by our method contain the more complete infrared targets and integrate more visible detail information.

### 4.7. Objective Evaluation

Since the visual sensitivity varies from person to person, subjective comparison analysis inevitably involves bias. Therefore, eight metrics and runtime are selected as the objective evaluation methods for evaluating the quality of fused images more convincingly and comprehensively. The proposed method and the other compared methods are analyzed quantitatively, and the results are shown in Figure 21.

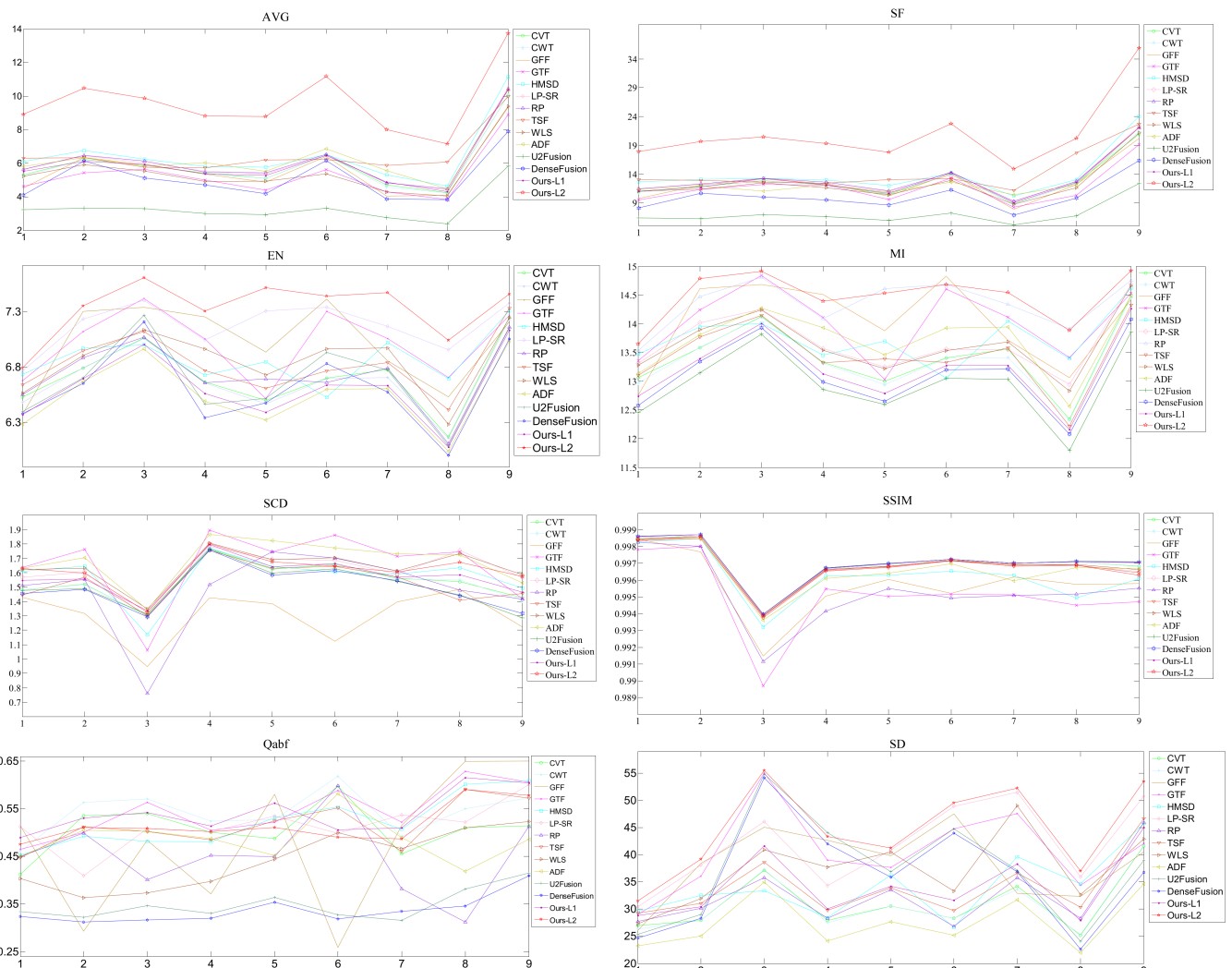

**Figure 21.** Quantitative comparison of eight objective evaluation metrics using the different fusion methods on the nine infrared and visible image pairs.

The quality of the fused images is in line with the values of the eight metrics. As seen from the summary of the various metrics in Figure 19, the proposed method with decomposition level 2 (L2) occupies a leading position on the metrics of AVG, SF and EN, which means that the proposed method L2 can retain more gradient information and has better performance in terms of the contrast and clarity of the images. The metric values of MI, SD, $Q_{abf}$, SCD and SSIM in the proposed method L2 are optimal or suboptimal for the nine groups of experiments.

In order to analyze these values comprehensively and directly, the average values of eight metrics in the nine groups of experiments are shown in Table 2. Each objective evaluation metric corresponding to a column and the best average value of every metric are marked in bold. From Table 2, the proposed method with decomposition level 2 shows

excellent performance in the cases of AVG, EN, MI, SF and SD. The other metrics infinitely approach the optimal value, such as SSIM, $Q_{abf}$ and SCD.

Generally speaking, the noises of images are prone to be calculated by $Q_{abf}$ and EN, which may lead to the higher and incorrect values. The metric EN is seriously affected by noises; for example, the results fused by the method RP have a lot of noises, especially in the *i*-th row of the seventh group experiment of Figure 19. Thus, the value of EN is the highest of all the methods. Moreover, the absolute-max rule is also able to choose the noise components as the final fusion coefficients. For instance, the GTF method adopts the absolute-max rule to guide the detail components to fuse, and $Q_{abf}$ is a bit higher. The more abundant the texture information is, the lower the $Q_{abf}$ is. The results of Our_L2 have more texture details than Our_L1, and the value of the metric $Q_{abf}$ was lower. The SD value of ADF is relatively the lowest, and the fusion results obtained by the methods are the most blurred from the subjective perspective. It can be seen from the last two rows that the fusion performance is improved with the increase in the decomposition level.

**Table 2.** The average values of the quality metrics for all source images.

| Metrics | AVG | EN | MI | $Q_{abf}$ | SCD | SSIM | SF | SD | Runtime/s |
|---|---|---|---|---|---|---|---|---|---|
| CVT | 5.93150 | 6.71624 | 13.43248 | 0.50564 | 1.54137 | 0.99696 | 12.71199 | 31.07864 | 2.56935 |
| CWT | 5.90104 | 6.69444 | 13.38888 | 0.54181 | 1.54074 | 0.99696 | 12.78876 | 30.91403 | 3.06967 |
| GFF | 5.64820 | 7.02521 | 14.05042 | 0.47950 | 1.30318 | 0.99573 | 12.42413 | 38.69633 | 3.76592 |
| GTF | 5.30221 | 7.01295 | 14.02590 | 0.54415 | 1.64860 | 0.99505 | 11.73354 | 40.60941 | 1.62658 |
| HMSD | 6.47369 | 6.87112 | 13.74224 | 0.52323 | 1.57335 | 0.99625 | 13.91654 | 34.09468 | 18.9567 |
| LP_SR | 6.15730 | 7.17781 | 14.35563 | 0.51370 | 1.47744 | 0.99530 | 13.21347 | 42.19765 | 0.76576 |
| RP | 6.49347 | 6.73163 | 13.46327 | 0.45079 | 1.53093 | 0.99679 | 14.28131 | 32.47274 | 0.58621 |
| TSF | 5.51254 | 6.82671 | 13.65343 | 0.51599 | 1.63624 | 0.99688 | 12.36487 | 33.97755 | 0.10834 |
| WLS | 6.30668 | 6.86250 | 13.72499 | 0.44229 | 1.67990 | 0.99660 | 12.70711 | 37.36021 | 4.77940 |
| ADF | 5.11189 | 6.55777 | 13.11554 | 0.48563 | 1.49885 | 0.99704 | 10.10662 | 27.60857 | 1.66422 |
| U2Fusion | 3.50888 | 6.67918 | 13.35836 | 0.33821 | 1.56602 | 0.97833 | 7.524127 | 36.39723 | 0.60190 |
| DenseFusion | 3.38556 | 6.48795 | 12.97590 | 0.33790 | 1.50824 | 0.99705 | 7.116488 | 26.79117 | 0.29560 |
| Our_L1 | 6.02180 | 6.61190 | 13.22379 | 0.50908 | 1.57302 | 0.99703 | 12.95345 | 34.19755 | 3.21583 |
| Our_L2 | 9.65432 | 7.20059 | 14.47895 | 0.49473 | 1.66344 | 0.99683 | 20.97932 | 43.56282 | 6.56182 |

From the results of the last column in Table 2, we can see that the runtime of various fusion methods is greatly different. Compared with the traditional methods, the deep learning methods have a significant advantage in terms of runtime due to the acceleration function of GPU. Moreover, the runtime based on the TSF method is the lowest because of the simple decomposition method and fusion strategy. Although the runtime of our proposed methods is a bit higher than that of the deep learning methods and some traditional methods, these are acceptable because of the better fusion effect. Of course, the promotion of the fusion efficiency will become our primary research aspect in the future.

In summary, the objective evaluation results are basically consistent with the subjective evaluation results. The proposed method (Our_L2) can fuse infrared and visible images in various scenes well. The fusion performance is the most stable and can integrate much more detail information into the final fused images. As a result, the fused images based on the proposed method have a better visual effect than the other compared methods.

## 5. Conclusions

In this paper, the proposed method based on LatLRR nested with RGF outperforms the other existing methods in terms of the fusion of infrared and visible images. The method first uses the LatLRR to extract the salient components from source images, and the detail components are processed further by RGF. Thus, the final base components can be obtained by the difference operator. Furthermore, the extracted detail components and the base components obtained by the difference are fused by the joint model between the nuclear-norm and modified spatial frequency and the l2-energy minimization model, respectively. Finally, the fused images can be obtained by linear summation operation.

The fusion performance of the proposed method is outstanding enough compared with the others, especially in terms of the image contrast and clarity, and so on. Moreover, the proposed method can adapt to various imaging scenes. In addition, nine pairs of infrared and visible images are selected to compare the fusion effect of our method and the other twelve methods. All the results are evaluated both objectively and subjectively, which demonstrates that the proposed method is superior to the existing methods in terms of the fusion performance.

**Author Contributions:** Conceptualization, B.Q.; methodology, B.Q.; validation, B.Q., H.L. and X.B.; investigation, B.Q. and Y.Z.; resources, H.L., G.L. and B.Q.; writing—original draft preparation, B.Q.; writing—review and editing, X.B. and W.W. All authors have read and agreed to the published version of the manuscript.

**Funding:** This work was supported in part by the National Natural Science Foundation of China under Grant 61801455.

**Data Availability Statement:** Not applicable.

**Conflicts of Interest:** The authors declare no conflict of interest.

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
