# Peer review of "A Novel Saliency-Based Decomposition Strategy for Infrared and Visible Image Fusion"

_remotesensing, doi:10.3390/rs15102624_

Round 1
Reviewer 1 Report
The infrared and visible image fusion algorithm designed in this paper from decomposition to fusion is reasonable and appropriate.The background of the research and relevant references are fully introduced in the introduction section.The article also fully describes the method and demonstrates the algorithm with a large amount of experimental data.The language of the article was used appropriately and no obvious errors were found.After careful reading of the article, the following comments are given.
(1) The article mentions the use of five sets of images to pretrain the projection matrix L in section 3.1.The specific method used to train L is not seen in the article. So I wonder if five sets of pictures are a bit less?
(2) When the LatLRR-RGF algorithm is used to decompose the image, the decomposition result obtained is used as input and decomposed, and so on repeatedly to finally obtain Ibi. Whether there is a definite method or judging criterion for the number of i.
(3) Is the threshold e used to smooth the image decomposed with a projection matrix?Why did it not appear in the previous algorithm introduction section, but suddenly appeared in the display of experimental results.
(4) Below the fourteenth formula, Ib should be the fused image.
I am very sorry for any error of the suggestions.
Author Response
Dear Professor:
On behalf of my co-authors, I sincerely thank the reviewer for giving me the opportunity and suggestions to revise and resubmit the manuscript.
We have already read reviewer’ reports carefully, and all comments have been accepted and counted. Necessary changes/corrections have been made to improve our manuscript. We revised the manuscript substantially. Please find the revised version and detailed list of the point-by-point response to comments.
We would like to express our sincere appreciation to you and reviewers for your patience and work on our manuscript. Please do not hesitate to contact us if there is anything, we can do for further quality promotion.
"Please see the attachment."
Yours sincerely,
Corresponding authors:
Biao Qi

Reviewer 2 Report
On the structure of the this paper:
1. it is rather long and should be preferably trimmed with emphasis in sections 1 and 2.
2. In section 4 there is a lot of discussion. Try to compress your presentation by aggregating your findings in some tables; e.g. method x vs performance in the presence of noise
3. While the presentation of relevant literature is comprehensive, I felt that the authors did not explore enough alternative method, such as pattern spectra, Characteristic scale- Saliency-Level (CSL) model based on Differential Area Profiles, and connected operators in general that are known to deliver crisp details of salient features in a strictly edge-preserving manner.
I am including some relevant citations should you wish to incorporate or discuss these domain in section 1 or 2.
- A new approach for the morphological segmentation of high-resolution satellite imagery by M. Pesaresi; J.A. Benediktsson
https://ieeexplore.ieee.org/abstract/document/905239
- The CSL model: https://www.spiedigitallibrary.org/conference-proceedings-of-spie/8390/1/A-new-compact-representation-of-morphological-profiles--report-on/10.1117/12.920291.short?SSO=1
Lastly, I have one minor request. This journal is on remote sensing and while the topic is very relevant there is no image demonstrating the proposed methodology on satellite imagery. You may easily access Sentinel2 images that contain a diverse set of frequency bands among which is optical and infrared (at the same resolution).
Please consider incorporating at least one such example in your analysis to make your work more relevant to this journal.
The level of English is good overall, yet some parts of the text require a bit of work. For example, articles are often missing making the text rather un-natural.
Below, I identified a few lines that would certainly need editing:
pp2, ln84: But this method is lack of spatial consistency and result in artifact effectiveness in the edges... -> But this method lacks spatial consistency and results in artifact effectiveness (???) in the edges...
pp5, ln203: And design two kinds of fusion rules for different layers, respectively. (???)
pp5, ln211: ...due to that the larger the image size, the more useful information it contains... -> ...due to that the larger the image size is, the more useful information it contains...
pp5, ln214: Besides, the setting of parameters n and the stride will be discussed in Section 4. -> The setting of parameters n and the stride will be discussed in Section 4.
pp18, ln508: In order to further verify the above analysis, in Figure 17, it shows that the results of nine -> To verify the above analysis further, Figure 17 shows that the results of nine...
pp18, ln511: An overall comparison of the different fusion results reveals as follows. (???)
pp20, ln569: In order to analysis these values -> In order to analyze these values
Author Response

(The authors gave the same response as above.)

Reviewer 3 Report
Dear authors,
The manuscript proposes a saliency-based decomposition strategy for infrared and visible image fusion.
The proposed description has several significant drawbacks that need to be considered:
1. There is no clear evidence that the proposed method outperforms existing fusion methods in terms of performance.
2. The manuscript is challenging to follow due to the use of various testing images for different presented tests, rather than using the same images throughout.
4. The source of the testing images is not adequately explained, nor is the rationale behind their selection.
5. the manuscript should have highlighted the geospatial context, which is missing, so major improvements are needed.
Specific comments:
The relationship between Figure 4, the supporting text, and equations 6-9 is not clear. Clarification is needed.
In Section 4.1, the rationale behind the selection of the comparison methods and the eight performance metrics should be justified to increase the clarity of the manuscript.
he meaning of the threshold "e" in Section 4.2 (Table 2) is not easy to understand. It is then later presented in Section 4.3.2. This should be explained in more detail.
Tables 1, 2, and 3 could be presented as charts, similar to Figure 12, to improve their clarity.
The meaning of the squares in the images in Figures 23 and 14 is not clear. This should be explained.
In Figure 16, it is unclear whether the "o" and "p" approaches are superior to the "e" and "g" approaches. Furthermore, Figure 17 is not detailed enough to compare different solutions. Additional information is needed to address these issues.
Author Response

(The authors gave the same response as above.)
